# Bee Bread Production—A New Source of Income for Beekeeping Farms?

**Piotr Semkiw *** **and Piotr Skubida**

The National Institute of Horticultural Research, Konstytucji 3 Maja 1/3, 96-100 Skierniewice, Poland; piotr.skubida@inhort.pl

* Correspondence: piotr.semkiw@inhort.pl

**Abstract:** Bee bread, i.e., floral pollen collected and partially processed by honey bees, is a source of many compounds beneficial for the human health. So far, the level of bee bread production in apiaries has been low due to many factors. However, development of such production may be significant as a new source of income for beekeepers. In spring 2015 a three-year study was started to determine bee bread production scale in honey bee colonies and assess the economic efficiency of such production. The experiment included 28 honey bee colonies each year; the colonies were divided into four groups. Each group tested different brood nest configuration or frames' placement against the hive entrance for the amount of harvested bee bread. All the costs, including labor input, were related to the process of bee bread production. Depending on the group, it was possible to harvest from 0.51 to 1.23 kg of bee bread from one colony. The average production amounted to 0.7 kg, and the entire apiary gave 20 kg of bee bread annually. Annual costs connected to bee bread production amounted to 679.5 EUR, while the estimated income from sales amounted to 1110 EUR. Thus, the profit was 430.5 EUR, i.e., 21.5 EUR per 1 kg of harvested bee bread. The highest costs were connected to labor and they may potentially comprise a factor limiting the development of bee bread production in apiaries.

**Keywords:** bee bread; beekeeping; beehive products; economic condition; production

---



## 1. Introduction

Honey is the main source of income in most of apiaries. Economic results of such production depend on many factors, e.g., scale of production (individual/global), honey variety or marketing strategy (selling at a purchasing center/directly to a customer). Furthermore, honey bee colonies' strength and survivability affected by disease, pesticides, and climate change influence the scale of production and consequently the economic results [1–4]. Recent years (not only in Poland) were not favorable to honey production as they negatively influenced economic condition of beekeeping, their competitiveness, and investment potential, along with high costs of apiary management [5–8]. That is why it is important to introduce additional sources of income to an apiary, for instance, through expanding its product range. Harvesting bee bread can be an interesting and lucrative solution although it is still little-known and scarce in apiaries. However, it is a very prospective solution, mainly because consumers' awareness in the area of the so-called functional foods (i.e., foods of beneficial and pro-health influence on human body, mainly due to natural content of biologically active compounds), is constantly growing [9]. There is no doubt that bee bread is an abundant source of such compounds because it has a rich composition of phenolic compounds and high content of flavonoids [10–14]. Aside from the issues related to the demand, while significant for the success of this undertaking, the development of bee bread production is also favored by market prices that are at least three times higher than those of honey [15]. Why then is the production of bee bread still marginal despite so many factors favoring such production? There are at least several

points to explain the issue. First of all, most beekeepers are convinced that the bee bread reserves of honey bee colonies should not be reduced in order to prevent suppressing colonies' development and lowering their biological condition [16,17]. Secondly, there is a lack of knowledge of the production volume, proper methods of managing honey bee colonies, harvesting dates, and potential costs [12,18]. While the first issue will be addressed in another paper because it requires referring to different aspects of honey bee biology, all the other issues will be discussed in this paper. Nevertheless, we would like to explicitly emphasize that we do not recommend this solution as an alternative to the production of honey but as an additional and concurrent method to the basic beekeeping business activity.

Due to the fact that harvesting bee bread is a relatively new topic, there is not a lot of literature data available on this subject. Previous studies focused mainly on collecting pollen (pollen loads) and their results usually indicated that in Polish climatic and flow conditions, without using any additional specific procedures, the level of pollen loads production per one honey bee colony amounts to about 2 kg. Implementing beekeeping technologies increasing pollen yield resulted in production of up to 8 kg per honey bee colony [19,20]. It was also reported that harvesting pollen was often the only product to obtain from weak colonies [21]. In the context of these studies, similar results can be expected when it comes to bee bread as pollen loads are the main material that honey bees need to produce bee bread. In the beginning of May, the weather and flow conditions are usually stable enough for the pollen inflow to the colonies, which are large enough to cover immediate needs of the colony and then store in combs as bee bread. Depending on the hive type and the beekeeping methods used, it is possible to differentiate several methods favoring storing bee bread in combs so that it is later possible to harvest it. In horizontal hives (common in small apiaries) directly behind the brood combs, an insulator and then a frame with light-brown comb is placed. After a few days it gets filled with freshly deposited bee bread. Then, this comb should be replaced again with a new comb so that the honey bees can further store the pollen. As the colony and the brood nest grows, the insulator and pollen combs are moved to the side (or to the back—depending on hive type and nest configuration) to make place for additional frames for brood rearing [16,17,22]. This method seems to be effective but is also very laborious and requires frequent interference in honey bee colonies. When the pollen flow is very abundant, isolating the brood nest with an insulator is usually unnecessary, because honey bees collect so much pollen that they have to store it in combs at the peripheries of the brood nest [16,17]. If possible, it is beneficial to arrange the brood nest in the so-called warm way, i.e., frames are parallel to the hive entrance. In this case, the queen is restricted to the combs at the back of the brood nest, and two–three combs closer to the entrance are used to store bee bread [23]. In modular hives, pollen is stored in combs located at the sides of the lowest hive body. However, in order not to restrict the queen from laying eggs too much, bee bread combs are moved to higher levels of the hive and replaced with light-colored combs or, when they are not available, with wax foundation frames [16]. In hives consisting of multiple modules, isolating the brood nest in the middle hive body, positively effects the amount of collected pollen. In this case, the lower hive body (under the first insulator) is used to store bee bread, and the higher one (above the second insulator) serves as the honey super [23]. It is worth noticing that bee bread combs can be taken out of the colony for future utilization at earliest after about two weeks from filling them with pollen loads which is necessary for the bee bread to ripen (fermentation process). Bee bread harvested earlier may have lower value and go bad easily [24]. For sanitary reasons one needs to remember that bee bread should be harvested from colonies free from any diseases.

Bee bread extraction from combs aiming at obtaining a product suitable for sale requires employing a technological process consisting of freezing the combs, then fragmenting them, separating beeswax and bee bread, drying the bee bread, and, finally, removing fine impurities. It is important to remove any residual honey from the combs with bee bread before starting the process [25]. Technically, the procedure of obtaining the final

product can be fully automated [26]. However, in typical apiaries, which are the most widespread apiaries in EU countries, where the average number of hives per beekeeper is 21 it is not economically justified [7]. In addition, to our knowledge, such technological lines are not yet produced. Most of the necessary works can be carried out using tools and devices already present in the apiary or household (freezer, pollen dryer, sieves with different mesh sizes). Additional device is only required for bee bread combs fragmentation and separation of wax and bee bread. Beekeeping equipment producers have been offering such devices dedicated for different apiaries already for several years. Prices range from about 150–200 EUR (for small and medium sized amateur apiaries—about 20–50 hives) to about 1500–2500 EUR (for large sized professional apiaries—more than 150 hives). Beekeeping internet fora and social media present different do-it-yourself methods of bee bread combs fragmentation, e.g., using a boring machine with a honey creaming screw. However, we would not recommend this method because it can cause serious damages to the product. Poland as well as EU lacks any normative documents concerning bee bread intended for sale. The only country having such regulations is Russia [27]. According to the Russian standard defining physicochemical parameters of bee bread, moisture content should not exceed 15%, residual wax should not exceed 5%, and the content of other mechanical impurities should not exceed 0.1%. Lowering bee bread water content to an amount indicated by this standard allows safe storage for 1 year at 0–15 °C.

The aim of his study was to assess potential bee bread production prospects and economic efficiency of such production in order to evaluate and disseminate this alternative production and source of income in Polish beekeeping farms.

## 2. Materials and Methods

The studies were conducted in three consecutive beekeeping seasons (2015–2017) in an experimental apiary of the National Research Institute of Horticulture, Apiculture Division in Puławy, Poland (51°24′29.7″ N 21°58′04.3″ E). Each year, the experiment included 28 colonies of *Apis mellifera caucasica*, kept in 12-frame Dadant hives with upper honey super ($\frac{1}{2}$ Dadant frame size). Apiary's was not stationary. The honey bee colonies were transported to various places to increase production. Before each winter they were moved to Sadłowice (51°23′01.3″ N 21°56′31.5″ E) and stayed there till every spring. This site provided honey bees with a flow of *Solidago* spp. and *Sinapis alba* nectar during autumn and mainly *Salix* spp. and *Acer* spp. nectar in spring. The same climatic and flow condition ensured similar development of honey bee colonies. At the beginning of rape blooming season (at the turn of April and May) the apiary was moved to Pulki (51°25′35.8″ N 22°06′37.5″ E) or Osiny (51°28′14.7″ N 22°03′14.4″ E). After the rape ceased blooming (about 25 May) the colonies were placed in Karczmiska Drugie (51°13′59.0″ N 22°00′19.5″ E) in raspberry (*Rubus idaeus*) fields. At the end of June, the apiary was moved to forest areas (fir stand) of the Suchedniów forest inspectorate (51°03′46.0″ N 20°41′24.7″ E) for honeydew flow. The main pollen sources for honey bees for bee bread production were: *Salix* spp., *Taraxacum officinale*, *Ribes nigrum*, *Prunus domestica*, *Malus domestica*, *Brassica napus var. oleifera* and *Rubus idaeus* during each season.

### 2.1. Standard Beekeeping Procedures

At the beginning of August, the apiary was moved to the wintering site. Then redundant combs were taken away from brood nest. Final arrangement of the nest and adjustment of honey bee colonies' strength to the number of combs left for winter was carried out at the beginning of September. At the same time, the honey bee colonies were treated against varroosis, winter feeding was done, and defective or weak honeybee queens were replaced with naturally mated egg laying ones (*Apis mellifera caucasica*, line Woźnica). To control the *Varroa destructor* infestation, Biowar 500, was used (500 mg of amitraz/strip, 2 strips per bee colony, exposure period—8 weeks). Control treatment after strips removal was performed with Apiwarol (12.5 mg of amitraz/tablet, 1 tablet per colony). Homemade saccharose syrup with beet sugar (sugar to water ratio 5:3) was used as a winter feeding

of the honey bee colonies. Each colony received about 15 kg of crystal sugar, expressed as dry mass. After winter feeding was terminated (September/October), feeders were removed from the hives. During routine inspections in spring, brood nests were extended, mainly using frames with wax foundation, and less frequently—frames with light-colored empty combs.

### 2.2. Preparing Experimental Groups

Experimental honey bee colonies were inspected at the middle of April of each year. This included assessment of brood amount and colonies' strength expressed as the number of combs covered by honey bees. Then 4 homogenous groups, each with 7 colonies, were created. Hive configurations in the groups were modified several days before moving the apiary to the rape flow site in order to create 4 different combinations to obtain bee bread combs:

1.  Group 1—symmetrical isolation of 7 brood combs with a queen bee with two vertical insulators; other combs, including combs with bee bread stored by honey bees earlier, are placed on both sides of the insulators (the insulator group).
2.  Group 2—isolation of 7 brood combs with a queen bee in the middle hive body with horizontal insulators; other combs, including combs with bee bread stored by honey bees earlier, are placed in an additional hive body under the brood nest (the hive body group).
3.  Group 3—frames arranged parallelly to hive's front wall (in the so-called warm way), applying the rule that bee bread combs are placed closest to the hive entrance, and brood combs are placed behind them (the warm way group).
4.  Group 4—standard arrangement of frames (perpendicularly to hive's front wall), however, brood combs are placed centrally, and other combs are placed in outmost locations of the hive body (the cold way group).

During the production period, additional honey supers were installed; one or two per hive depending on flow intensity and amount of honey gathered by honey bees.

### 2.3. The Process of Bee Bread Extraction

Bee bread combs were collected from the colonies just before moving the apiary to the honeydew flow site. Then, the initial state of frames arrangement was also restored. Bee bread combs (Figure 1a) were marked to enable later identification (colonies/groups) and then frozen. Bee bread was extracted from the frozen combs, then dried, purified, and weighed in order to determine amounts of bee bread harvested from individual colonies and groups. Beeswax and bee bread were separated mechanically using a device of high efficiency, commercially manufactured for this purpose. The device operates in a following way: crashing bee bread combs frozen at $-18$ °C (without damaging bee bread), then winnowing beeswax and small impurities two times (Figure 1b).

Bee bread was dried at 39 °C for about 3 days in a dryer used typically for pollen loads drying (Figure 1c).

The next process, i.e., final purification, involved using a sieve with mesh size slightly smaller than bee bread cells and, when necessary, removing remaining impurities and cocoon residues manually. Purified product was weighed, packed, labelled, and intended for sale (Figure 1d).

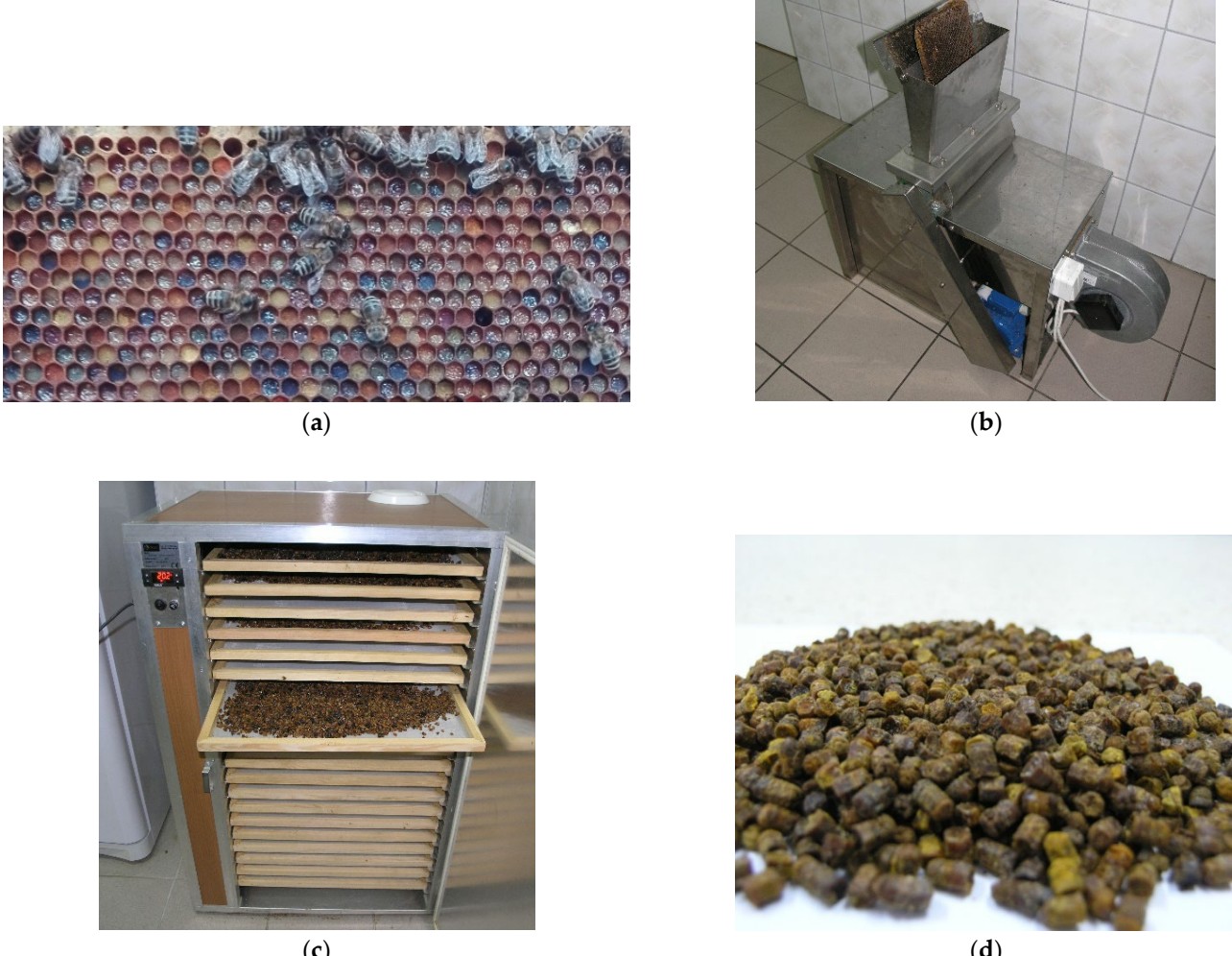

**Figure 1.** (**a**) Fragment of bee bread comb; (**b**) bee bread harvesting machine; (**c**) bee pollen dryer machine; (**d**) purified bee bread—ready for sale.

### 2.4. Statistical Analysis

Statistical calculations were carried out using the Statistica v.10 software (2011). Numerical data were subjected to distribution analysis using the Kolmogorov-Smirnov test (K-S) with Lilliefors correction (K-S-L) (Figure 2). Normal data distribution had brood area (Kolmogorov-Smirnov test $D = 0.059$, Lilliefors test $p > 0.20$) and bee bread production (Kolmogorov-Smirnov test $D = 0.05$, Lilliefors test $p > 0.20$. Abnormal data distribution had number of combs covered by bees (Kolmogorov-Smirnov test $D = 0.27$, Lilliefors test $p < 0.01$) and bee bread area (Kolmogorov-Smirnov test $D = 0.14145$, Lilliefors test $p < 0.01$). Data with abnormal distribution confirmed by the analysis underwent log transformation (10). Finally, based on the obtained results, to analyze data concerning bee bread mass and brood area in colonies, one-way ANOVA with Duncan post-hoc test was used, whereas colony strength and bee bread area were analyzed using the Kruskal-Wallis median non-parametric test with multiple comparisons. For the assessment of interactions, the two-way ANOVA with Duncan post-hoc test was sed. The level of significance was set at $\alpha = 0.05$.

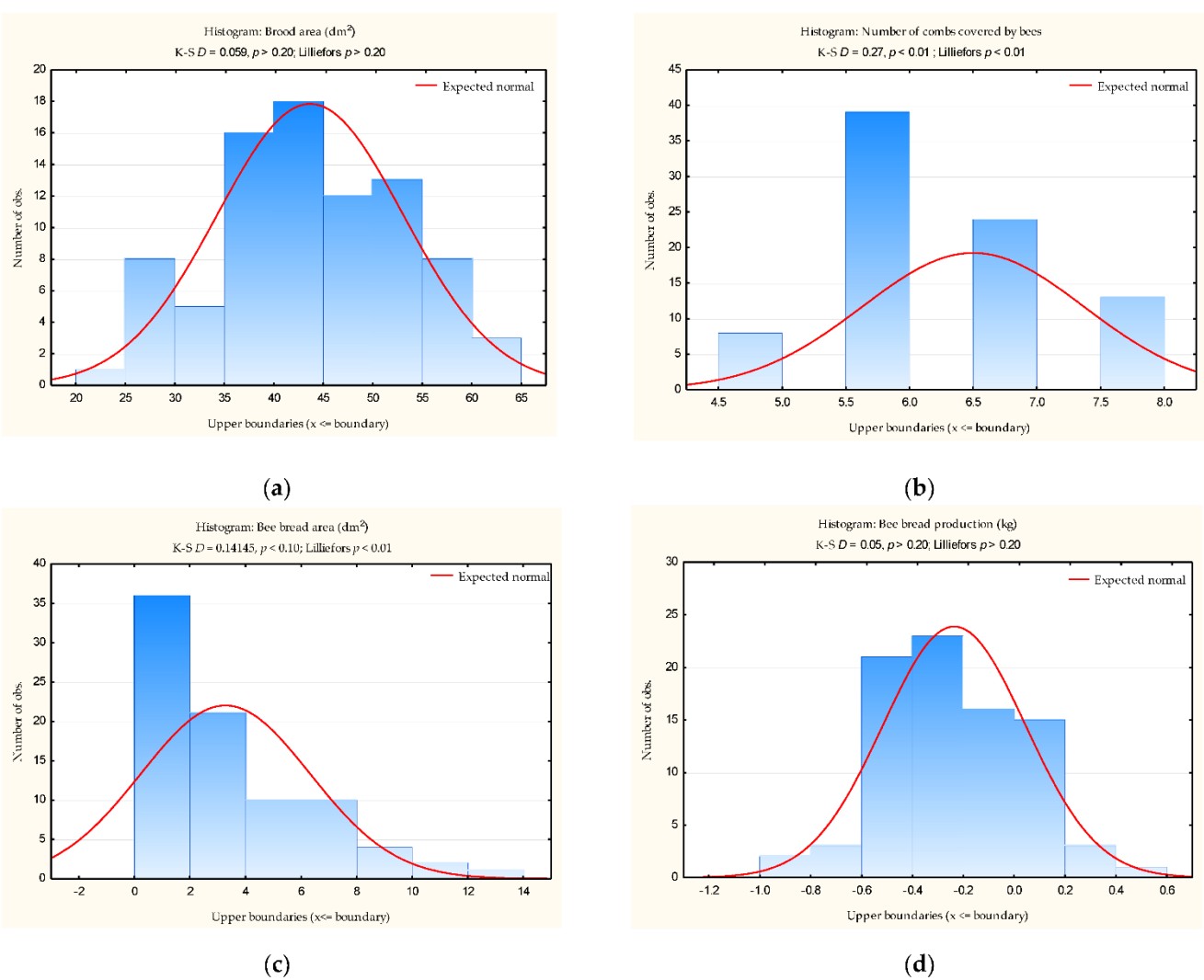

**Figure 2.** Data distribution: (**a**) brood area (dm$^2$); (**b**) number of combs covered by honey bees; (**c**) bee bread area (dm$^2$); (**d**) bee bread production (kg).

*2.5. Economic Calculations*

Costs of producing 1 kg of bee bread were estimated assuming average annual production of 20 kg and taking into consideration the following components: straight line depreciation of the devices, labor, electric energy and travel costs. PLN is the official currency of Poland, however, for the purpose of this elaboration all amounts were expressed in EUR, assuming mean exchange rate of 4.50 PLN [28]. In the case of devices, the annual depreciation amount equaled 10%. The labor input was calculated registering all the activities performer in the apiary connected solely to obtaining bee bread. Cost of 1 worker-hour (wh) was determined as 4.07 EUR and the cost of 1 kWh of electric energy—as 0.1475 EUR [29,30]. Travel costs were estimated based on the total distance travelled, average fuel consumption of the used vehicle (9l/100 km), and average fuel price of 1.09 EUR/liter [31]. Furthermore, costs of packaging and labelling were estimated at 1.5 EUR/kg. The cost of the combs was excluded because re-used usually for 2/3 years for the production of the bee bread or for "regular" comb brood or honey. The average price of 1 kg of bee bread in retail sale was determined based on the analysis of prices of this product in 14 Polish online and stationary shops offering bee products. We discussed a price of bee bread with 6 beekeepers that sell the product directly from the farm. The prices for a product weighing from 500 to 1000 g were taken into account. Based on above-described analysis, the average bee bread retail price was estimated at 55.5 EUR/kg. Total income

was calculated based on annual production and sale of 20 kg of bee bread at the price of 55.5 EUR/kg, which was estimated on the analysis of prices of this product. Total costs connected to bee bread production was calculated based on particular fixed and variable costs, resulting from our experience.

## 3. Results

### 3.1. Production of Bee Bread in the Honey Bee Colonies

The study included 28 colonies each year, thus, in the three-year study period, the total of 84 colonies were included in the experiment. At the stage of creating experimental groups, the colonies had similar brood area, number of combs covered by honey bees, and bee bread stores (Table 1). These colonies were in good biological condition, had no clinical symptoms of any diseases, and were of optimal, for the spring, production potential. The groups did not differ significantly between years in terms of: brood area ($dm^2$)— $F_{(6,72)} = 0.13825$, $p = 0.83574$; number of combs covered by honey bees—$H_{(3,84)} = 0.5913751$, $p = 0.8984$; bee bread area ($dm^2$)—$H_{(3,84)} = 0.4016395$, $p = 0.9399$.

**Table 1.** Biological status of honey bee colonies at preparation of experimental groups.

| Year of Study | Group (*n*) | Brood Area ($dm^2$) Mean $\pm$ SD [1] | No. of Combs Covered by Honey Bees Mean $\pm$ SD | Bee Bread Area ($dm^2$) Mean $\pm$ SD |
|---|---|---|---|---|
| | Insulator (7) | 43.7 $\pm$ 11.5 | 6.6 $\pm$ 1.0 | 3.0 $\pm$ 2.2 |
| | Hive body (7) | 42.5 $\pm$ 9.0 | 6.4 $\pm$ 1.0 | 3.5 $\pm$ 2.1 |
| 2015 | Warm way (7) | 43.6 $\pm$ 9.6 | 6.4 $\pm$ 1.0 | 3.4 $\pm$ 2.5 |
| | Cold way (7) | 42.2 $\pm$ 8.4 | 6.6 $\pm$ 0.8 | 3.3 $\pm$ 1.5 |
| | *p*-value | 0.78 [2] | 0.89 [3] | 0.86 [3] |
| | Insulator (7) | 47.6 $\pm$ 9.9 | 6.6 $\pm$ 0.8 | 2.8 $\pm$ 3.1 |
| | Hive body 7) | 47.0 $\pm$ 7.6 | 6.6 $\pm$ 0.8 | 3.2 $\pm$ 4.5 |
| 2016 | Warm way (7) | 47.5 $\pm$ 7.8 | 7.0 $\pm$ 0.8 | 3.0 $\pm$ 3.4 |
| | Cold way (7) | 47.5 $\pm$ 12.1 | 6.9 $\pm$ 0.9 | 3.3 $\pm$ 1.5 |
| | *p*-value | 0.69 [2] | 0.92 [3] | 0.91 [3] |
| | Insulator (7) | 41.3 $\pm$ 9.2 | 6.4 $\pm$ 1.0 | 3.6 $\pm$ 3.8 |
| | Hive body (7) | 40.6 $\pm$ 11.9 | 6.1 $\pm$ 1.1 | 3.2 $\pm$ 4.5 |
| 2017 | Warm way (7) | 40.7 $\pm$ 6.5 | 6.3 $\pm$ 1.0 | 3.9 $\pm$ 4.4 |
| | Cold way (7) | 40.0 $\pm$ 9.9 | 6.1 $\pm$ 0.7 | 3.0 $\pm$ 2.9 |
| | *p*-value | 0.88 [2] | 0.90 [3] | 0.82 [3] |

[1] Standard deviations; [2] ANOVA test; [3] Kruskal-Wallis H test; $\alpha = 0.05$.

In total, during the experiment, 60.3 kg of bee bread was obtained from honey bee colonies (Table 2.). The average production per unit amounted to 0.72 kg, with a quite wide range of values (from 0.1 to 2.78 kg). The average amount of bee bread obtained from one colony did not differ significantly between years, however, a trend of higher production in 2015 was noticed, especially in comparison to 2017. This resulted in bee bread production higher by 6 kg in total. The average amount of annual production from the entire apiary (all the experimental groups) was 20 kg.

In 2015, no significant differences in bee bread production were stated between experimental groups, despite noticeably higher values in the hive body group (Table 3). In this group, 1.1 kg of bee bread was harvested per group, on average, while the average production in other groups reached only about 0.7 kg/colony. A year later, differences in production levels in individual groups were higher and statistically significant ($F_{(3,24)} = 6.8363$, $p = 0.00172$). The highest amount of bee bread was obtained from the hive body group—1.5 kg/colony, and the lowest in the warm way group—0.35 kg/colony. The average production in the remaining groups was slightly over 0.5 kg/colony. It was striking that the total amount of bee bread obtained from the hive body group—10.6 kg,

was higher than the cumulative production from the remaining 3 groups—10.5 kg. In 2017, differences between groups were also significant ($F_{(3,24)}$ = 5.3499, *p* = 0.00577). The highest amount of the product was obtained from the hive body group—1 kg/colony, whereas in the remaining groups the level of production was lover by half or more.

**Table 2.** Production of bee bread during 3 years of study.

| Year of Study | No. of Honey Bee Colonies | Bee Bread Production (Kg) | | | |
|---|---|---|---|---|---|
| | | Mean | Range (Min–Max) | SD | Total |
| 2015 | 28 | 0.81a | 0.15–2.28 | 0.45 | 22.7 |
| 2016 | 28 | 0.75a | 0.10–2.78 | 0.64 | 21.1 |
| 2017 | 28 | 0.59a | 0.19–1.91 | 0.41 | 16.5 |
| Total | 84 | 0.72 | 0.10–2.78 | 0.51 | 60.3 |

Means followed by the same letters do not differ significantly, *p* = 0.24. SD—standard deviation.

**Table 3.** Production of bee bread depending on individual groups and the years of study.

| Year of Study | Group (*n*) | Bee Bread Production (Kg) | | | |
|---|---|---|---|---|---|
| | | Mean [1] | Range (Min–Max) | SD | Total |
| | Insulator (7) | 0.69a | 0.37–1.35 | 0.33 | 4.8 |
| | Hive body (7) | 1.12a | 0.34–2.28 | 0.63 | 7.9 |
| 2015 | Warm way (7) | 0.74a | 0.15–1.43 | 0.44 | 5.2 |
| | Cold way (7) | 0.69a | 0.31–0.98 | 0.23 | 4.8 |
| | *p*-value | 0.39 | | | |
| | Insulator (7) | 0.59a | 0.10–1.32 | 0.38 | 4.1 |
| | Hive body (7) | 1.52b | 0.37–2.78 | 0.80 | 10.6 |
| 2016 | Warm way (7) | 0.35a | 0.21–0.5 | 0.11 | 2.5 |
| | Cold way (7) | 0.55a | 0.28–1.15 | 0.28 | 3.9 |
| | *p*-value | 0.002 | | | |
| | Insulator (7) | 0.48a | 0.29–0.98 | 0.24 | 3.4 |
| | Hive body (7) | 1.05b | 0.25–1.91 | 0.55 | 7.3 |
| 2017 | Warm way (7) | 0.44a | 0.26–0.63 | 0.14 | 3.1 |
| | Cold way (7) | 0.38a | 0.19–0.79 | 0.2 | 2.7 |
| | *p*-value | 0.006 | | | |

[1] Means marked by the different letters indicate significant differences at α = 0.05. SD—Standard Deviation.

During the three-year study, 21 repetitions for each experimental group were conducted (Table 4). The significant differences were found between groups ($F_{(3,72)}$ = 10.999, *p* = 0.0000). The highest bee bread production per honey bee colony was found in the hive body group and amounted to 1.23 kg/colony, on average. Level of bee bread production of 0.6 kg per one colony was found in the group with the brood nest isolated with two insulators, and in the case of the warm way and cold way group, on average, 0.5 kg of bee bread was harvested. About 43% of the total amount of bee bread harvested from all of the colonies originated from the hive body group. The analysis of interaction between study years and the amount of bee bread harvested in individual groups did not show statistically significant differences ($F_{(6,72)}$ = 1.0522, *p* = 0.39930) (Figure 3).

**Table 4.** Production of bee bread in the studied group—total for all years.

| Group | No. of Honey Bee Colonies | Bee Bread Production (Kg) | | | |
|---|---|---|---|---|---|
| | | Mean [1] | Range (Min–Max) | SD | Total |
| Insulator | 21 | 0.59a | 0.1–1.35 | 0.32 | 12.3 |
| Hive body | 21 | 1.23b | 0.25–2.78 | 0.67 | 25.8 |
| Warm way | 21 | 0.51a | 0.15–1.43 | 0.31 | 10.8 |
| Cold way | 21 | 0.54a | 0.19–1.15 | 0.26 | 11.4 |

[1] Means marked by the different letters indicate significant differences at α = 0.05. SD—standard deviation.

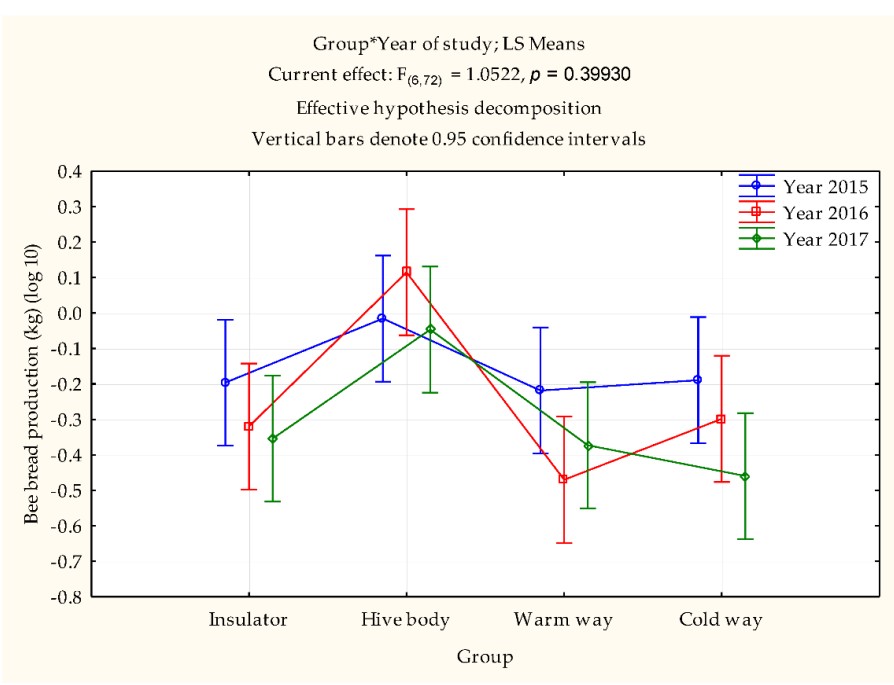

**Figure 3.** Relationship between the year of study, tested group and bee bread production.

### 3.2. Calculations of the Economic Efficiency of Bee Bread Production

Annual costs of bee bread production amounted to 649.5 EUR (Table 5). In their structure, fixed costs and variable costs were distinguished. Among fixed costs straight line depreciation of the devices used in the process of production was registered. In the case of the device for comb fragmentation and separation of beeswax and bee bread the straight-line depreciation was 140 EUR/year, and in the case of the pollen dryer—25 EUR/year (the total annual costs of depreciation amounted to 50 EUR, however, as the device is also used to dry pollen loads, half of this amount was included in the calculations). The variable costs included costs of electric energy powering the devices, labor costs, and travel costs. Packaging and labelling costs were also included. The total energy consumption connected to the operation of devices (freezer, bee bread harvester, and bee pollen dryer) and lighting the workplace amounted to 60 kWh. All the activities carried out in order to prepare the final product (honey bee colonies management, collecting bee bread combs, operating devices, manual separation of bee bread, etc.) took 112 worker-hours. Thus, the total labor costs equaled 456 EUR. The total annual milage of travels to the apiary where bee bread was harvested amounted to 200 km. Based on the fuel consumption of the used vehicle and fuel oil unit price, the travel costs amounted to 19.5 EUR. Costs of packaging and labelling 20 kg of beebread amounted to 30.0 EUR. In the general costs structure, the fixed costs comprised 24.3% and the variable costs—75.7%. Out of all the costs, the labor costs were of highest share (67.1%).

Annual production and sale of 20 kg of bee bread at the price of 55.5 EUR/kg generated total income of 1110 EUR (Table 6). After deducting all the costs, the total net profit amounted to 430.5 EUR and the net profit per unit—21.5 EUR/kg.

**Table 5.** The average annual costs connected to bee bread production.

| Cost Breakdown Structure | | Quantity | Total Amount (EUR) | Percentage Share |
|---|---|---|---|---|
| Fixed costs | Linear depreciation of the: | | | |
| | -bee bread harvester | 10% per year | 140 | 20.6 |
| | -bee pollen dryer | $\frac{1}{2} \times 10\%$ per year | 25 | 3.7 |
| | Total fixed costs | | 165 | 24.3 |
| Variable costs | Electricity | 60 (kW-h) | 9 | 1.3 |
| | Worker-hours | 112 (wh) | 456 | 67.1 |
| | Transportation | 200 (km) | 19.5 | 2.9 |
| | Packaging and labelling | 20 (kg) | 30 | 4.4 |
| | Total variable costs | | 484.5 | 74.6 |
| Total cost (fixed + variable) | | | 679.5 | 100 |

**Table 6.** Annual profitability analysis of bee bread production.

| Components | Value (EUR) |
|---|---|
| A—total cost | 679.5 |
| B—total gross income | 1110 |
| C—total net income (B–A) | 430.5 |
| D—gross income of 1 honey bee colonies (B/28) | 39.6 |
| E—net income of 1 honey bee colonies (C/28) | 15.4 |
| F—net income of 1 kg of bee bread (C/20) | 21.5 |

## 4. Discussion

The studies were conducted in an apiary that, despite its experimental status, was intended for honey production. That is why the conditions of performing the experiment were similar to the conditions in commercial apiaries. The experiment was organized in such a way to find out the possibilities of bee bread production in honey bee colonies that are moved to different flow sites during season, both nectar and honeydew. This guaranteed that obtained results will have an interesting implementation value, and on the other side, the results can serve as a base of objective economic calculations. This was important because no detailed data on this issue were so far present in the literature. Previous information was dispersed and rather general [16,17,22,23]. Thus, our study will undoubtedly enrich existing knowledge that is important for the development of this area of beekeeping. It was proved that planned bee bread production is possible and that bee bread can be obtained from regular, not only defective combs. Moreover, necessary economic calculations were conducted to assess profitability of such production. Bee bread combs were removed from the colonies at the end of June, after the dynamic spring development, replacement of the winter generation of honey bees, and reaching optimal condition by honey bee colonies. It was expected that such procedure would not negatively influence on functioning the honey bee colonies and would not disrupt honey production. It needs to be mentioned that measurements allowing to assess the influence of harvesting bee bread on biological and production parameters were also analyzed. However, the obtained results will be reported in a separate paper.

The study tested several different combinations of managing colonies in relation to the most effective production of bee bread for adequately long time (three beekeeping seasons). All applied combinations allowed to obtain bee bread combs and the average bee bread production amounted to 0.72 kg/colony. However, it turned out that the best configuration to increase the amount of the obtained product was the one using additional hive body, conventionally called the hive body group. The results obtained for this group particularized information available in the literature [23]. In contrast to the source data, results in the group where the brood nest and other frames were arranged in the so-called warm way did not meet the assumed expectations. Bee bread production in those colonies

was at the same level as the production in the group with standard arrangement of beehive frames, i.e., the cold way group. It can be assumed that the low yield resulted from the fact that only the combs with bee bread were removed from the colonies. Combs with both bee bread and brood, even if in small amounts, were not taken out. Different procedure could have resulted in much higher production. Isolating the brood nest with two insulators in the insulator group enabled to obtain slightly higher amounts of bee bread in comparison to the warm way and cold way groups, however, this was below the expectations and the difference was so slight and not confirmed statistically. Maybe it would be possible to obtain better results if the brood nest would be isolated on smaller number of combs. Concern of limiting the space for the queen bee to lay eggs too much resulted in not making any corrections, the more so because maintaining optimal strength of honey bees was significant in the context of utilizing the honeydew flow. This made the honey bees store high amounts of pollen in the nest combs between the insulators. Moreover, especially in this group it was noticeable that pollen loads that were just gathered where stored in a form of chains surrounding the brood in order to provide the young honey bees with proper food. This supports the results of studies on honeybee preferences concerning foraging on bee bread conducted by Carrol et al. [31]. These authors systematically monitored stores and consumption of this food and showed that honey bees distinctly prefer to eat freshly stored pollen (storing period of about two to four days) despite having quite large amounts of previously gathered stores of protein food. The cage studies conducted in this experiment showed that fresh (one-day bee bread) was chosen by honey bees three times more frequently in comparison to bee bread stored in the hive for 10 days. Five-day product was consumed two times less frequently than the one-day product. It is worth mentioning that the results obtained in this study contradict the previously recognized hypothesis that only well conserved pollen is a valuable nutrient for honey bees. Based on the cited report, it can be assumed that the logistics of honey bees in relation to storing bee bread around brood cells significantly limits the possibility of harvesting this product. Only high pollen flow results in storing bee bread in other combs and such situation is basically the main chance to obtain satisfactory production. Managing honey bee colonies in a way enabling the highest possible harvest of bee bread is a secondary matter which is also illustrated by a graph presented in Figure 3. According to this graph, irrespective of the year, i.e., also of the amount of potential pollen collection as a material to produce bee bread, the amounts of this product obtained in individual groups were similar. It can be also added that to provide proper honey bee colonies' functioning during and after harvesting of bee bread, restrictive methods limiting laying eggs by queens should not be used.

Another aspect of harvesting bee bread should be also mentioned, i.e., such production can also be an element of refreshing combs in honey bee colonies. This is because all combs with bee bread are removed from the hive and replaced with wax foundation frames or light-colored beeswax frames. In the hive body method due to high number of combs that can be removed this process can be carried out on a higher scale.

Under the experimental conditions the average production amounted to 0.72 kg/colony. However, it needs to be emphasized that the reported amounts concerned bee bread that is dry, purified, and prepared for sale. At this level of production and current retail prices, the possible income from one honey bee colony is about 40 EUR. In comparison to other bee products this corresponds to 3.6 kg of pollen loads (average price for 1 kg—11 EUR) or 6 kg of multifloral honey (average price for 1 kg—6.7 EUR) [5]. If as the reference, for instance for honey, the wholesale price for 1 kg is assumed (2.9 EUR/kg) then the amount of honey needed to match the income from selling bee bread is 13.8 kg. The calculations of economic efficiency of bee bread production included only costs closely related to this production. Other costs of managing honey bee colonies were not taken into account because, as it has been mentioned before, the bee bread production is recommended only as an additional income, accompanying the main business activity, i.e., honey production. Labor costs had the highest share in variable costs and they had the highest impact on the profitability of

production. The highest amount of labor was needed to carefully separate bee bread from all residual wax and cocoon. This also persuaded authors to search the methods to produce bee bread using artificial combs that would allow to skip many activities usually needed in the process of production [18]. However, it needs to be clearly stated that the product obtained this way cannot be called a natural bee bread and that it is only its unfortunate copy. The method employing artificial combs should not be promoted among beekeepers due to low quality of the final product that only for a layperson would remind its natural counterpart. A better method to reduce labor related to bee bread production would be taking care of the quality of the combs in honey bee colonies. Preparing bee bread for sale takes a lot less time when it is harvested from combs that have not gone through too many brood rearing cycles. Light-brown combs are the most suitable for this purpose. Bee bread harvested from such combs does not contain a lot of impurities that would make the process of purification harder. In addition, this way obtained product will not be as loaded with different residues of acaricides used to fight *Varroa*.

It is worth to mention that currently bee bread has limitation of which most important are scale of production and labor cost. Therefore, research in this direction is developmental, so in the future, ways of increasing production scale together with reducing labor costs should be sought.

## 5. Conclusions

Bee bread production in honey bee colonies is possible and its scale depends on the intensity of pollen flow and employed method of managing honey bee colonies. Described research and results set out a method for beekeepers potentially interested in this area of business. Comparison of costs and incomes resulting from bee bread production showed satisfactory financial effects. Labor costs might comprise a factor limiting the development of harvesting this bee product. Nevertheless, it is good to engage in this area of production because bee bread has a beneficial impact on human immunological system, antibiotic and antioxidative properties, and it is better assimilated by the human body in comparison to floral pollen. In the era of different civilization diseases bee bread can be a valuable product that the consumers frequently look for—this should additionally encourage beekeepers to harvest bee bread. Moreover, from the economic point of view, production of bee bread could be an alternative and effective form of diversifying the beekeeping activity increasingly threatened by a number of issues and increase beekeepers' incomes. Further research on this subject should be conducted for refining the production technique, make the supply chain more efficient and assess consumer appreciation.

**Author Contributions:** Conceptualization and methodology, P.S. (Piotr Semkiw), P.S. (Piotr Skubida); investigation, P.S. (Piotr Semkiw), P.S. (Piotr Skubida); writing—original draft preparation, P.S. (Piotr Semkiw), P.S. (Piotr Skubida); writing—review and editing, P.S. (Piotr Semkiw), P.S. (Piotr Skubida); visualization, P.S. (Piotr Semkiw), P.S. (Piotr Skubida); supervision, P.S. (Piotr Semkiw); All authors have read and agreed to the published version of the manuscript.

**Funding:** This research received no external funding.

**Institutional Review Board Statement:** Not applicable.

**Informed Consent Statement:** Not applicable.

**Data Availability Statement:** Not applicable.

**Conflicts of Interest:** The authors declare no conflict of interest.

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
