# Peer review of "Bee Bread Production—A New Source of Income for Beekeeping Farms?"

_agriculture, doi:10.3390/agriculture11060468_

Round 1

Reviewer 1 Report

I thank the authors and the editors for giving me the opportunity to review this manuscript. I have learned a lot about bee bread, a topic which is not often addressed in the literature. I think the topic is of interest and I praise the authors for a nice integration of beekeeping and economic analysis in one well articulated study.

The paper is well presented in terms of structure and figure but the English is quite difficult to read and must be revised by a native speaker in order to give any chance of wide readership to the article. This is a fairly easy thing to accomplish I believe.

I do have more substantial concerns. One about the methodology and one about the tone of the paper. 

On the methodology, I think that it should be emphasised that the sample size is very small and many of the parameters are just single point estimates. The result tables with standard deviations give the impression of a much more robust statistical analysis but in fact, with such a small number of observations, it is hard to believe that any statistical effects should be trusted as reliable.

A further point on the methodology relates to the economic data. There is only one price point of 55 euros per kg for bee bread and I could not find where the "market analysis" for that price comes from. I would suspect that bee bread is a niche product and therefore a lot more effort needs to be undertaken to document the potential for expanded sales. Figures for the cost side also would benefit from a more careful documentation. The authors do point out that labor may be a limiting factor in the development of the activity, which is a good start. An interesting question from there is whether that is the reason why bee bread has not become a more widespread product if it is indeed profitable.

This leads to the tone of the paper. As is, the paper reads like a pamphlet from someone in the bee bread industry. The claim is that bee bread production is profitable and given the thin evidence on the price side and the very small sample side, it hard to consider the claim to be supported by scientific evidence. 

My criticism is not about any purported intentions of the authors or about the interest of the topic, of whether bee bread is a profitable activity. 

I hope the authors will decide to strengthen the economic analysis with a more thorough price data search and revise the manuscript as I find the topic to have potential interest for many readers.

Author Response

The authors thank the reviewer for valuable comments and suggestions.

Reviewer 2 Report

Dear authors, 

the submitted manuscript is written at a very high level and brings interesting insight into bee bread production. As I am a beekeeper I evaluate the content of the paper as very significant and original. Methodology and results are clearly described. The overall merit of the paper is very high.

However, I suggest including in the Discussion section also study limitations if applicable and mentioning what should be addressed in the future studies

Author Response

The authors thank the reviewer for valuable comments and suggestions.

Point 1: However, I suggest including in the Discussion section also study limitations if applicable and mentioning what should be addressed in the future studies

 Response 1: We added information to the text.

The authors thank the reviewer for valuable comments and suggestions.

Reviewer 3 Report

Dear Authors, the manuscript is interesting but you have to implement the methods referred to economic calculations and revise the calculations of the economic efficiency.  Best regards

In the text you should replace everywhere:

-beebread with bee bread (cfr Eva Crane)

-bee(s) with honey bee(s),

-bee colonies with honey bee colonies or colonies (better)

Line 23 add beehive products as keyword

Line 29 before Recent years,…added something that link economic results to colonies threats. For example: Furthermore, honey bee colonies strength and survival affected by disease, pesticides, climate change [Goulson, D.; Nicholls, E.; Botías, C.; Rotheray, E.L. Bee declines driven by combined stress from parasites, pesticides, and lack of flowers. Science 2015, 347, 1255957; Flores, J.F.; Gil-Lebrero, S.; Gámiz, V.; Rodríguez, M.I.; Ortiz, M.A.; Quiles, F.J. Effect of the climate change on honey bee colonies in a temperate Mediterranean zone assessed through remote hive weight monitoring system in conjunction with exhaustive colonies assessment. Sci. Total Environ. 2019, 653, 1111–1119; Calatayud-Vernich, P.; Calatayud, F.; Simó, E.; Pascual Aguilar, J.A.; Picó, Y. A two-year monitoring of pesticide hazard in-hive: High honey bee mortality rates during insecticide poisoning episodes in apiaries located near agricultural settings. Chemosphere 2019, 232, 471–480; Vercelli, M.; Novelli, S.; Ferrazzi, P.; Lentini, G.; Ferracini, C. A Qualitative Analysis of Beekeepers’ Perceptions and Farm Management Adaptations to the Impact of Climate Change on Honey Bees. Insects 2021, 12, 228. https://doi.org/10.3390/insects12030228] influence the production of market goods and consequently the economic results.

Line 30 economic condition of apiaries bee farming system

Line 31 [1,…] add other references referred to EU situation such as:

  • European Commission. Deloitte. Evaluation of Measures for the Apiculture Sector. Final Report; Publications Office of the EU: 496 Luxembourg, 2014.
  • European Union. Honey. Detailed Information on Honey Production, National Apiculture Programmes, Budget and Legal Bases. 2021. Available online: https://ec.europa.eu/info/food-farming-fisheries/animals-and-animal-products/animal-products/honey_en

and economic papers such as

Mancuso, T.; Croce, L.; Vercelli, M. Total brood removal and other biotechniques for the sustainable control of Varroa mites in honey bee colonies: Economic impact in beekeeping farm case studies in Northwestern Italy. Sustainability 2020, 12, 2302.

Line 49 honey bees’ colonies’ biology

Line 98 specify what do you mean by typical apiaries? Specify please

Line 104 small apiaries (specify the number of beehives please), the same for large apiaries

Line 115-116 implement the text “…in order to..alternative production, source of income…as indicated in the title…in Poland…”

Line 122 why location is not fixed? The apiary was not in your research institute? Since the other coordinates are indicated, it would be better to indicate them or specify the reason for absence

Line 125 the flow guarantees only the development of the colonies or honey production?

Line 129 delete L. after Rubus idaeus or put L. in other scientific names in all the text such as Sinapis alba, Varroa destructor

Line 118-131 since the manuscript deals with bee bread derived by pollen of plant species, indicate some scientific name of species as done for the nectar flow. Even if you don’t analyse the pollen I think you know the flora present near the apiaries involved in the research and the relative bloomings.

Line 138 add “,” after Biowar 500

Line 141 sugar syrup? In winter the sugar is given as candy. specify better your preparation, please

Line 154 …strength expressed add “in this case or in our experiment” as the number….because other methods exist

Page 195 you should to combine the figures with a single caption (a, b,…) or at least put them side by side to improve the graphics

Page 195 after the process, cited the parameters indicated in statistical analysis as listed in fig.5: brood area (dm2), number of combs….in the 4 different groups were calculated….and the methods to obtain (cite the equations for costs, income etc.)

Line 210 you have to underline that the cost of the combs is excluded because re-used usually for…years for the production of the bee bread or for “normal” comb: brood, honey, pollen

Line 219 if you used the retail prices you have to consider not only the production costs but also the packaging and sell. If you indicate the wholesale prices you only have to change table 6 rather than making changes to the total costs but I understand that this price not exist because the production is rare. You can’t compare costs and revenues referred to different levels.

Line 220 add something about the income calculation reported as results in table 6.

Line 275 You don’t have to consider fixed cost (otherwise calculate all the allocated costs) (Table 5) but only variable cost adding within the variable cost linked to commercialization of the product (see rev line 219).

So, you could recalculate the variable costs including all supply chain and modify table 5 and 6 and the text.

In table 6 only we will deal with total gross income and not total net incomes (no fixed costs calculated).

Revise results and discussion (the main text is good) on the basis of the above calculation.

In conclusion section, put more emphasis on the bee bread production as new/alternative/innovative source of income in bee farm system.

Author Response

(The authors gave the same response as above.)

Round 2

Reviewer 1 Report

I thank the authors for the detailed answers to my first report.

The addition of method details about the sourcing of price data is very useful and gives the paper a much more robust footing in arguing that there might be indeed revenue and profit opportunities for Polish beekeepers.

Author Response

We sincerely appreciate all valuable comments and suggestions, that helped us to improve the quality of the paper.

Best regards

Piotr Semkiw and Piotr Skubida

Reviewer 3 Report

Dear Authors, the manuscript is ameliorate. I attach my last comments to improve the text.

  1. About the Title I suggest (sorry, I don’t know how in the old revision this comment was deleted)

Bee bread production—a new source of income for apiaries beekeeping farms (or bee farm or bee farm system)?

The process involves apiaries but also machineries etc. present in honey extraction laboratory. It is better to consider all the step of the process as chain of beekeeping farm.

I suggest turning the title into a question, it is more appealing. Also, to affirm that it is cheaper in absolute terms you have to consider market supply/demand, etc. In my opinion, it’s a good compromise.

In the abstract it’s ok because you used may ….”However, development of such production may be significant as a new source of income for beekeepers.”

And in the paragraph lines 342-344: The aim of his study was to assess potential bee bread production prospects and economic efficiency of such production in order to evaluate and disseminate this alternative production and source of income in Polish apiaries beekeeping farms/…/..

Also, in the conclusion if you want to try to highlight the response to the title question.

  1. Line 362: The main pollen sources for honey bees for bee bread production were plants such as:
  2. 1380….form of diversifying the beekeeping activity increasingly threatened by a number of issues and increase beekeepers’ income.

I propose to insert a sentence such as (change as you prefer): further research on this subject will be conducted for refining the production technique, make the supply chain more efficient and assess consumer appreciation.

So, you will also support the question in the title.

This may be a good idea to the next step of the research. Good luck!

Best regard

Author Response

(The authors gave the same response as above.)
